# Mapping recurrent mosaic copy number variation in human neurons

Chen Sun [1,23], Kunal Kathuria [2,23], Sarah B. Emery[3], ByungJun Kim[1], Ian E. Burbulis [4,5], Joo Heon Shin [2], Brain Somatic Mosaicism Network*, Daniel R. Weinberger [2,6,7], John V. Moran [3,8], Jeffrey M. Kidd [1,3], Ryan E. Mills [1,3] ✉ & Michael J. McConnell [2] ✉

When somatic cells acquire complex karyotypes, they often are removed by the immune system. Mutant somatic cells that evade immune surveillance can lead to cancer. Neurons with complex karyotypes arise during neurotypical brain development, but neurons are almost never the origin of brain cancers. Instead, somatic mutations in neurons can bring about neurodevelopmental disorders, and contribute to the polygenic landscape of neuropsychiatric and neurodegenerative disease. A subset of human neurons harbors idiosyncratic copy number variants (CNVs, "CNV neurons"), but previous analyses of CNV neurons are limited by relatively small sample sizes. Here, we develop an allele-based validation approach, SCOVAL, to corroborate or reject read-depth based CNV calls in single human neurons. We apply this approach to 2,125 frontal cortical neurons from a neurotypical human brain. SCOVAL identifies 226 CNV neurons, which include a subclass of 65 CNV neurons with highly aberrant karyotypes containing whole or substantial losses on multiple chromosomes. Moreover, we find that CNV location appears to be nonrandom. Recurrent regions of neuronal genome rearrangement contain fewer, but longer, genes.

It is inaccurate to view an individual's genome as invariant from organ to organ, or from cell to cell within an organ. For example, somatic mosaicism among lymphocytes has been recognized since the 1970's with the discovery of somatic gene rearrangement at T cell receptor and immunoglobulin loci[1]. In the late 90's, advances such as spectral karyotyping (SKY)[2] and multiplex fluorescence in situ hybridization (FISH)[3] began to comprehensively map aneuploidy and chromosomal translocations in metaphase spreads from cancer cells. These approaches identified recurrent chromosomal translocations in proliferative cancer cells[4] leading, in part, to the identification of genomic fragile sites that underlie the ontogeny of many cancers[5]. When applied to neural genomes, SKY and FISH detected aneuploid neurons[6–8]. Recent advances in single cell and bulk DNA sequencing approaches have revealed abundant somatic mosaicism throughout the human body[9–13]. Associated studies have linked environmental mutagens to somatic mutations in the skin, bladder, and other

[1]Department of Computational Medicine and Bioinformatics, University of Michigan Medical School, 100 Washtenaw Avenue, Ann Arbor, MI 48109, USA. [2]Lieber Institute for Brain Development, 855 North Wolfe Street, Baltimore, MD 21205, USA. [3]Department of Human Genetics, University of Michigan Medical School, 1241 East Catherine Street, Ann Arbor, MI 48109, USA. [4]Department of Biochemistry and Molecular Genetics, University of Virginia, School of Medicine, Charlottesville, VA 22902, USA. [5]Facultad de Medicina y Ciencia, Universidad San Sebastián, Sede de la Patagonia, Puerto Montt, Chile. [6]Department of Psychiatry and Behavioral Sciences and Neuroscience, Johns Hopkins School of Medicine, 600 North Wolfe Street, Baltimore, MD 21287, USA. [7]McKusick-Nathans Institute of Genetic Medicine, Johns Hopkins School of Medicine, 733 North Broadway, Baltimore, MD 21230, USA. [8]Department of Internal Medicine, University of Michigan Medical School, 1500 East Medical Center Drive, Ann Arbor, MI 48109, USA. [23]These authors contributed equally: Chen Sun, Kunal Kathuria.*A list of authors and their affiliations appears at the end of the paper. ✉e-mail: remills@umich.edu; mikemc@libd.org

exposed cells[12,14,15]. Rapidly dividing stem cell populations also incur somatic mutations due to DNA replication errors. Clonal expansion of variant genomes can, in turn, shape mosaicism among an individual's somatic cells[16]. Somatic mutations, accompanied by cell death, set the stage for somatic selection during the lifespan of an individual.

Brain somatic mosaicism is associated with neurodevelopmental disorders, especially epilepsy[17–24]. Unlike other organs, cerebral cortical neurons arise *in utero* and are not replaced during normal human lifespan[25]. Neural stem and progenitor cells proliferate rapidly during human cortical development; these progeny overpopulate the developing cerebral cortex[26–29]. Somatic selection is one means by which some progeny may thrive as cortical neurons while other progeny succumb to neurodevelopmental cell death. The genomes of mature cortical neurons contain hundreds of single nucleotide variants (SNVs), some of which mark clonal lineages[25–28]. Long INterspersed Element-1 (LINE-1) mobile elements retrotranspose during neurogenesis and contribute to brain somatic mosaicism in a subset of neurons[30–35]. Although SNVs are numerous and accumulate throughout life, relatively few are predicted to cause protein-coding mutations with obvious consequences for affected neurons[36,37]. Megabase (Mb)-scale copy number variants (CNVs) - typically sub-chromosomal deletions—also contribute to brain somatic mosaicism[38–40].

In non-diseased (neurotypical) brains, dozens of genes are impacted in CNV neurons with substantial inter-individual variation in the frequency of CNV neurons. CNV neurons are more prevalent in the frontal cortex of young individuals ($n = 4$ individuals <30 years old; 28.5% CNV neurons, 75/263) than in aged individuals ($n = 5$ individuals >70 years old; 7.3% CNV neurons, 26/354)[41]. However, the small sample sizes in previous studies (<100 neurons/individual)[38–41] have limited power to find recurrent patterns of genome rearrangement (i.e., CNV hotspots) in any single individual. If present, recurrent sites of neuronal genome rearrangement could be influenced by common chromosomal fragile sites that are predisposed to genome rearrangements[42,43] or emerge via neurodevelopmental somatic selection. Neither mechanism is mutually exclusive.

Here, we show that recurrent brain CNVs occur during an individual's development, moreover hotspots and cold spots for CNV location are found among neurons in one individual's frontal cortex. A commercial droplet-based whole genome amplification (WGA) method was used to generate Illumina sequencing libraries from 2125 frontal cortical neuronal nuclei from a previously characterized neurotypical individual[37,41]. Read-depth analysis of each library is coupled with phased germline single nucleotide polymorphisms (SNPs) to develop a single cell Sequencing COVerage and ALlele-based approach (SCOVAL) that filters read-depth based deletion calls using concordant, phased, loss-of-heterozygosity (LOH) information. In total, 2097 single neuron libraries pass quality controls (QC) and 10.8% (226/2097) contain at least one Mb-scale CNV. An unexpected subpopulation of these CNV neurons (65/226, 25%) have highly aberrant karyotypes wherein multiple chromosomes harbor multiple deletions, including six aneusomic neurons. When compared to a random model, CNVs are depleted in gene-dense genomic regions. However, neuronal genome rearrangements are more common in genomic regions that contain genes encoded by more than 100 kilobases (kb) of genomic sequence (herein defined as "long" genes).

## Results

### Determining the genetic architecture of individual neurons

Whole and sub-chromosomal CNVs have been reported in human neurons by several previous studies that used three different WGA approaches (degenerate oligonucleotide-primed(DOP)-PCR[38–40], StrandSeq[44], or Picoplex[41]) followed by short read sequencing of pooled single nucleus libraries. Each laboratory assessed 20 to 120 frontal cortical neurons in different individuals, and all WGA approaches identified CNV neurons. Here, we applied a fourth WGA approach

(10X Genomics Single Cell CNV) that uses droplet-based microfluidics to enable the analysis of hundreds to thousands of single nuclei from a sample. In this approach, WGA is performed on thousands of nuclei, each individually encapsulated in a hydrogel. Hydrogel beads retain amplified genomic DNA, and are then microfluidically paired with barcodes, leading to a library pool containing hundreds to thousands of single nuclei.

We isolated neuronal nuclei from postmortem frontal cortex of a 49-year-old, male, neurotypical control by fluorescence-activated nuclei sorting. Using NeuN-positive nuclei (Supplementary Fig. 1A), two DNA libraries were prepared in separate lanes on the 10X Genomics Chromium platform (Fig. 1A); each lane produced ~1000 single neuronal genomic libraries with unique barcodes. The resultant libraries (2125 total) were combined into one pool, which was sequenced in two batches on an Illumina NovaSeq platform, achieving an average of $2.83 \pm 1.22$ million reads per neuron. Following our previous approach[41], we mapped reads to 5067 variable-sized autosomal bins, each containing 500 kb of uniquely mappable sequence (mean bin size = 569 kb, range = 501 to 2812 kb). Our quality control (QC) filters excluded 28 single neurons with aberrant bin-to-bin variance [i.e., median absolute deviation (MAD), 2097 (>95%) libraries passed QC] and masked 308 genomic bins that were outliers in global read coverage across all neurons (Supplementary Fig. 1B–D). We adapted Ginkgo[45] to call CNVs larger than 1 Mb, defined copy number (CN) state thresholds (see Methods), and identified 2564 putative autosomal CNVs (2401 deletions and 163 duplications) in 469 different neurons (Fig. 1B and Supplementary Data 1).

To develop SCOVAL, we performed 10X Genomics linked-read sequencing[46] on dural fibroblast DNA from the same individual at high coverage (~52.7X). This approach enabled the identification and phasing of germline SNPs by isolating long DNA segments into barcoded short reads that could be used to reconstruct underlying haplotypes into 2548 phased genomic blocks (mean 1178 kb ± 2034 kb, median 234 kb, max 17.15 Mb). Within each of these phased blocks, we further segmented the genome into windows of 20–100 phased heterozygous germline SNPs (mean = 107 kb, range = 0.687 to 1470 kb) that were used to arbitrate predicted somatic deletions with phased LOH. For each window of each cell, we counted the number of informative reads (e.g., reads that intersect with phased heterozygous SNPs) on each haplotype. We then calculated the absolute log2 ratio of the number of reads on each haplotype and integrated this ratio into the filtering models (Fig. 1C). The application of our naïve Bayesian-based pipeline (see Methods, Supplementary Fig. 2) identified 1985 regions with both sequence coverage and phased LOH support consistent with heterozygous deletions in 231 neurons. We excluded Gingko deletion calls where more than 75% of internal phased SNP windows contained fewer than three informative reads and arrived at a call set of 1853 heterozygous somatic deletions in 226 neurons.

SCOVAL produced a final deletion CNV set (Supplementary Data 2) comprising 1957 somatic CNV calls (13.95 Mb ± 17.47 Mb) among 226 CNV neurons (~11%). These represent 76.3% of the initial 2564 read-depth predictions. Notably, CNV neuron prevalence (226/2097 neurons) using droplet-based WGA (10X) and SCOVAL is in good agreement with previous read-depth based CNV detection using an alternative WGA approach (Picoplex) from this individual (~11%; 11 of 99 neurons)[41]. Although the nature of single-cell DNA sequencing prohibits the direct validation of identified CNVs, manual, subjective inspection of read-depth and allele ratios are strikingly concordant.

Other candidate neuronal CNVs (i.e., duplications and homozygous deletions) were more challenging to validate using SCOVAL. Previous studies using read-depth alone reported more than two-fold fewer duplications than deletions[41]. Using SCOVAL, we measured allelic ratios between haplotypes to assess the 163 Ginkgo duplication calls. The log2 ratios of haplotype-resolved alleles for each duplication were not significantly different from randomly sampled euploid

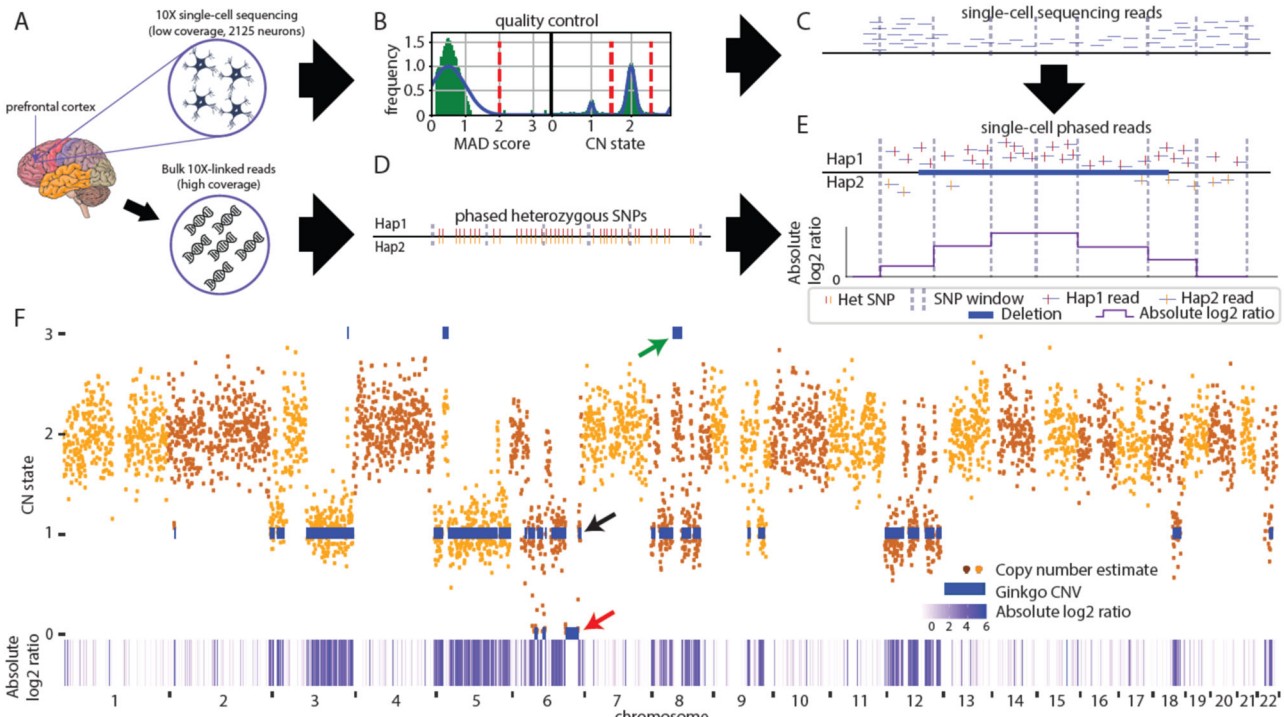

**Fig. 1 | SCOVAL: identification of copy number variation using read-depth and allele imbalance.** Overview of SCOVAL. **A** Single nuclei and bulk dural fibroblast DNA were analyzed using 10X platforms. (Images from vecteezy.com) **B** Single nuclei library quality is assessed based on median absolute deviation (MAD) and copy number thresholds are established using population statistics. Graphs depict schematized data; vertical red lines illustrate threshold strategy. **C** Candidate CNVs are identified based on altered read depth across consecutive genomic bins. **D** Heterozygous SNPs are phased using bulk linked-reads in chromosomal segments ("hap 1" or "hap 2"). **E** Absolute log2 ratios derived from "hap 1"/"hap 2" are calculated across ~100 SNP windows (see text). A deletion with concordant loss of heterozygosity (log2 ratio <> 0) is illustrated. **F** A highly aberrant CNV neuron (#5) shows representative Gingko calls (blue bars), duplications (e.g., green arrow), heterozygous deletions (e.g., black arrow), and homozygous deletions (e.g., orange arrow) and qualitatively concordant increases in absolute log2 ratio (white<purple). The genome is plotted from left to right on the x-axis, read-depth is in the upper panel (CN state on the Y-axis), and absolute log2 ratios are reported in the lower panel.

regions of that particular cell (one-tailed $t$-test, $p$ value = 0.998, Supplementary Fig. 3A). These findings suggest that greater single cell sequencing coverage likely will be required for SCOVAL to assess duplications in single neuron WGA data, although phased LOH may also allow us to filter regions where Ginkgo reports false positives (Fig. 1F, green arrow). Nevertheless, although some of these regions may represent bona fide duplications, we opted to exclude putative duplications with only Gingko support from further analysis in the interest of evaluating a conservative call set.

Homozygous deletions have been uncommon in previous datasets and have distinct properties compared to heterozygous deletions. Specifically, these deletions are not directly amenable to allelic modeling as both haplotypes are absent, and any observed non-zero allele ratios likely would be derived from mis-mapped reads. Thus, we developed an additional filter to reduce the false positive rate for 106 putative homozygous deletions with read-depth support. We calculated a read-depth ratio for each Ginkgo window by comparing the read-depth in every cell with the read-depth from bulk sequencing[37] and derived a Gaussian mixture model to calculate the posterior probability for putative homozygous deletions using these values from our initial heterozygous and homozygous deletion calls (see Methods, Supplementary Fig. 3B) This strategy found additional support for 86/106 putative homozygous deletions (posterior probability >0.99, Supplementary Fig. 3C). These 86 regions were included in our final deletion call set for subsequent analyses of CNV locations. Importantly, homozygous deletions are only found in neurons with highly aberrant karyotypes and all flank a heterozygous deletion (Fig. 1F, red arrow), indicating that they are likely the result of two independent and overlapping heterozygous deletions. Further, we identified 8

Ginkgo-called homozygous deletions that exhibited a read-depth and allele ratio profile consistent with heterozygous deletions and reclassified them as such (Supplementary Fig. 4).

We next assessed whether any of our somatic CNVs could potentially represent germline variants that escaped our analytical filters. We first examined the 10X linked-read data and called CNVs using LongRanger and Manta (see Methods). We did not observe any events larger than 1 Mbp nor any that had any considerable overlap with our somatic CNVs. We next examined the minor allele frequencies of heterozygous SNPs across all cells within the coordinates of our somatic CNVs and observed a median minor allele frequency (MAF) ranging from 0.45–0.49 (Supplementary Fig. 2D), consistent with typical diploid regions. Additionally, our detection resolution of >1 Mbp suggests that such events in the germline could presumably be pathogenic and thus are unlikely, given that the donor was healthy at the time of death. In aggregate, these results, coupled with the pathogenicity of such large CNVs, suggest that the presence of germline CNVs in our somatic set is unlikely.

SCOVAL was designed to identify idiosyncratic CNVs in human neurons. Another single-cell CNV caller, CHISEL, was designed to study tumor evolution and intra-tumor heterogeneity[47]. CHISEL and similar approaches[48] assume a higher frequency of tumor subclones (>5–10%)[49] than has been observed in CNV neurons[41]. When we tested CHISEL using our single neuron data, almost all reported CNVs (21,906) clustered collectively within 12 genomic loci (99.25% of CHISEL calls) and were reported in more than 50% of neurons (Supplementary Fig. 5). Notably, 11 of the 12 loci overlapped with SCOVAL outlier bins that were associated with WGA artifacts (see Methods and ref. 41). We next compared the remaining 165/21906 CHISEL CNV calls

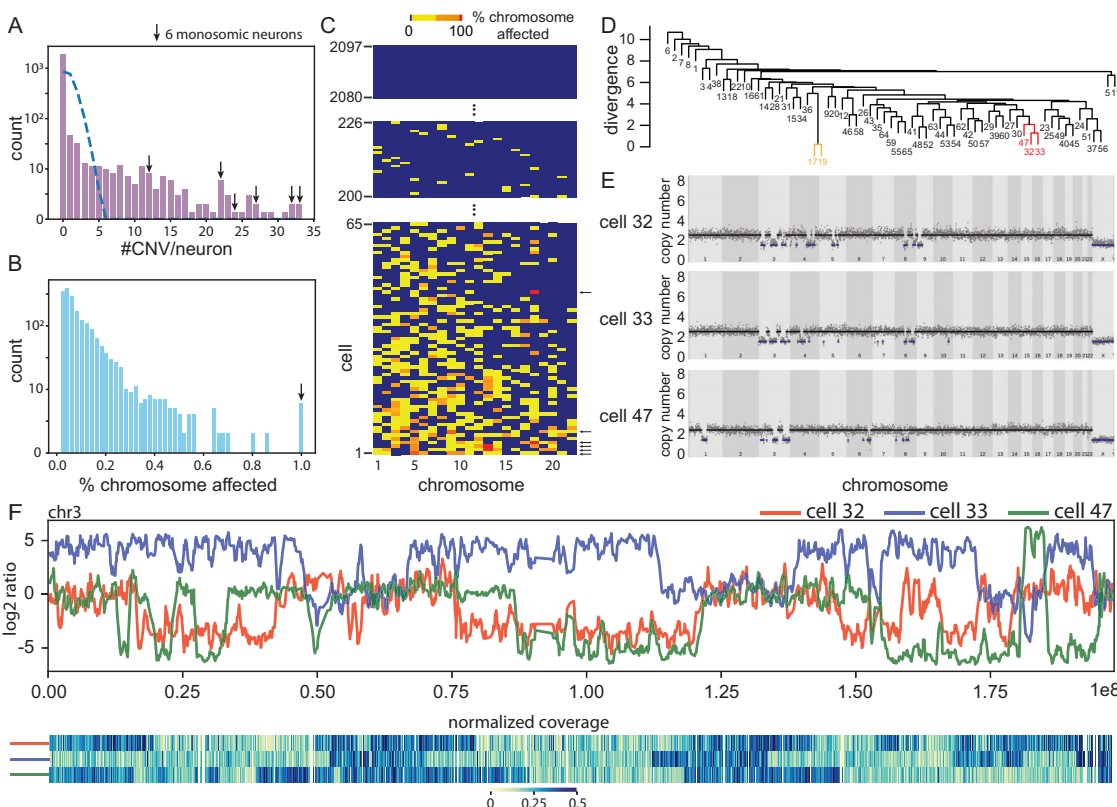

**Fig. 2 | CNV neurons can have highly aberrant karyotypes. A** The observed CNV per neuron [(purple bars, counts (y-axis),CNVs/neuron (x-axis)] distribution deviates (*P* < 0.0001) from Poisson expectations (dashed blue line). Arrows indicate neurons with monosomic chromosomes. **B, C** Deletions cluster in a subset of CNV neurons. **B** Counts (y-axis) of the cumulative percent of each chromosome deleted (*n* = 2097 neurons * 22 autosomes) in CNV neurons. **C** Neuronal genomes (*n* = 2097) are arranged in a cells-by-chromosome matrix, ranked by the total percentage of their genome containing deletions. Cell #226 is the first CNV neuron among 2097 total neurons with the smallest observed single deletion (blue = unaffected chromosome, yellow <50%, orange = 50−99%, red 100%). **D–F** Among 65 neurons with the most aberrant genomes, some have similar karyotypes. **D** Hierarchical clustering identifies two groups (yellow, red) with the least divergence from similarity (y-axis). **E** Red cluster neurons [cells #32, 33, and 47 in (**C**)] have similar CNV profiles. Read-depth is plotted as in Fig. 1F. The yellow cluster (cells #17 and #19) is shown in Fig. S6B. **F** Concordant read-depth is observed on opposite haplotypes in the most similar pair [#32(red) and #33(blue)]. When overlapping, events on cell #47 (green) match the #32 haplotype, but never the #33 haplotype. Chromosome 3 is plotted from left to right. Haplotype log2 ratio (upper panel) and corresponding read-depth (lower panel, blue = diploid) plots show overlapping deletions and LOH for each haplotype.

with our final call set. These 165 calls were reported in only three neurons, but 39 CHISEL CNV calls overlapped with 15 SCOVAL CNV calls. Manual inspection of read-depth and LOH at the other 126 CHISEL CNV calls found no subjective support (Supplementary Data 3). Consistent with reports attempting to apply similar cancer-oriented approaches for identifying somatic CNVs in neurons[37], approaches that rely on clonal information do not appear to be appropriate to study brain somatic mosaicism.

**Some CNV Neurons have highly aberrant karyotypes**

SCOVAL identified 226 CNV neurons with at least one deletion. These deletions ranged in size from 1Mb to whole chromosome losses (i.e., aneusomy). We also observed that when neurons harbored multiple deletions, many clustered on single chromosomes. In contrast to a uniform background model (see Methods and below), CNVs did not appear to be distributed randomly among CNV neurons (Fig. 2A). Forty-six CNV neurons contained a single deletion, but five contained greater than 30 deletions. Apparent chromosomal monosomies (i.e., where all genomic bins reported a copy number (CN) state = 1) were observed in six different neurons. One neuron (#1) was monosomic for Chr5, another neuron (#7) was monosomic for Chr9, two neurons (#2, #3) were monosomic for Chr13, and two other neurons (#4, #46) were monosomic for Chr18 (Fig. 2B, C). All monosomic neuronal genomes were highly aberrant and harbored many additional deletions affecting 40−98% of other chromosomes (Fig. 2C and Supplementary Fig. 6A).

Among 65 CNV neurons with deletions affecting >5% of their genome, 48 contained at least one chromosome that was >50% monosomic.

We evaluated CNV locations in CNV neurons based on the percentage of each chromosome affected by CNVs (Fig. 2C) and found two pairs of neurons (#17, #19 and #154, #155) that were nearly identical in their genomic read-depth patterns and could, in principle, represent clonal "sister" neurons that arose from a common progenitor cell during neurodevelopment (Supplementary Fig. 6B, C). However, each of these pairs arose from the same 10X Genomics Chromium lane; therefore, we cannot exclude the possibility that one nucleus may have paired with two barcodes in a single droplet. Subsequent analyses assume that these two pairs are highly concordant technical replicates.

Hierarchical clustering (Fig. 2D) identified three other neurons (cells #32, #33, and #47) with similar karyotypes that could, in principle, share identity by descent (Fig. 2E). Thus, we investigated whether these deletions occurred on the same chromosomal phase block (i.e., haplotype). Multiple deletions in cells #32, #33, and #47 mapped to Chr3; however, read-depth alone cannot assess whether these deletions occur on the same physical chromosome.

Linked-read sequencing identified Mb-scale phase blocks. To determine phasing at a chromosome level, we generated extended phase blocks using three CNV neurons (cells #33, #10, and #5) that contained overlapping deletions accounting for the full-length of Chr3 (Supplementary Fig. 7). Although CNV locations overlapped among

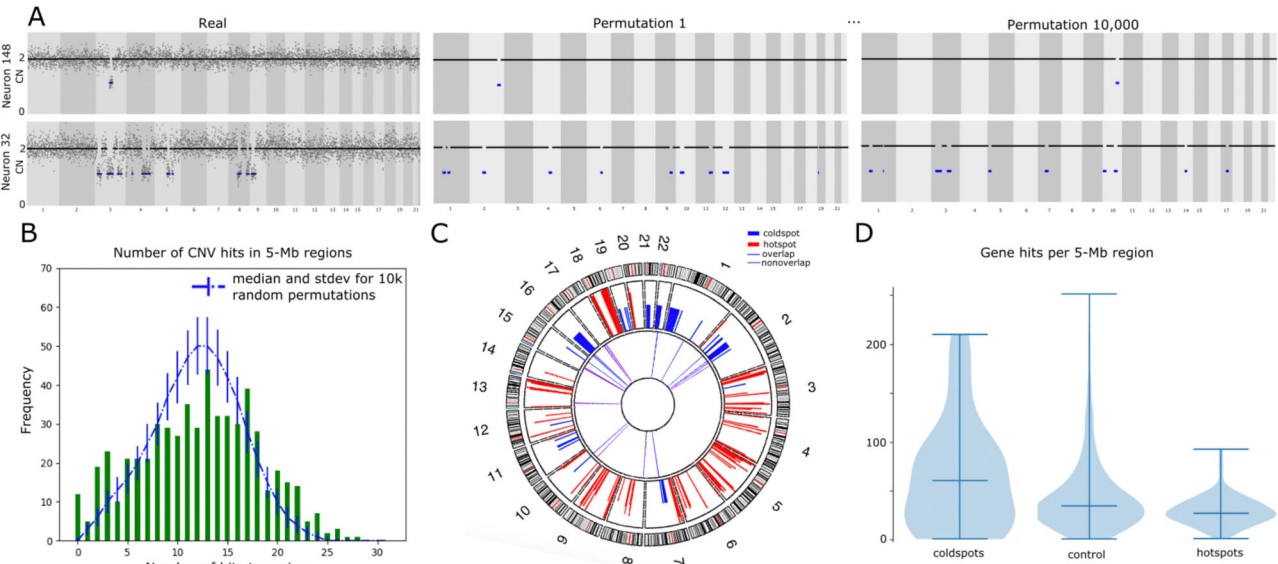

**Fig. 3 | Analysis of CNV distribution relative to random null model. A** Empirical read-depth plots of two CNV neurons (left panels) and representative permutations (right two panels) are displayed as in Fig. 1F. **B** Relative to 10,000 permutations of real data (represented by blue dotted line and error bars), high and low CNV burden are enriched at the extremities of the Gaussian distribution (green bars). **C** Circos plot shows that hotspots (red, outer tier) and cold spots (blue, outer tier) cluster on distinct chromosomes. Thirty-three pathogenic CNVs (blue, purple, inner tier) never overlap hotspots. Eleven (blue) overlap cold spots. **D** Violin plot (mean +/- SD) showing gene enrichment in cold spots (left, $N = 56$, mean gene hits = $57.71 \pm 58.93$) and depletion in hotspots (right, $N = 83$, mean gene hits = $15.65 \pm 32.40$) relative to other 5 Mb regions ($N = 404$ controls, mean gene hits $37.79 \pm 43.71$) with chi-square $P < 0.001$ for cold spots vs. controls and hotspots vs. controls.

these three neurons (Fig. 2F), the Chr3 CNVs were constrained to one haplotype in two neurons (cells #32 and #47), but occurred on the other haplotype in the third neuron (cell #33). The presence of other idiosyncratic CNVs suggest that these three neurons arose in distinct neurodevelopmental lineages. The possible ontogeny of these chromosomes might include chromosome mis-segregation, micronucleus formation, and a chromothripsis-like event[50–54]. In any case, the strikingly similar patterns of loss observed in these three neurons likely represent recurrent rather than clonal events.

## CNVs are not randomly distributed in neuronal genomes

The similar patterns of chromosomal loss observed in subsets of CNV neurons led us to hypothesize that, in contrast to what has been reported in other tissue types[55], neuronal CNV locations may not arise randomly. Thus, we generated a control dataset of randomly placed deletions and explored whether neuronal genomes accumulate CNVs in "hotspots" or are protected from CNVs in "cold spots." Briefly, the empirical call set was randomly rearranged, without collision, while keeping the size and abundance of CNVs constant on a per-neuron basis. We reasoned that randomly, and reiteratively, placing the "real" CNVs throughout the genome would effectively generate a "random" CNV landscape (Fig. 3A); thus, we performed 10,000 synthetic iterations of real data to generate a null model. For analysis, the genome was segregated into 567 contiguous 5 Mb regions and the number of simulated CNVs that overlapped each 5 Mb genomic region (i.e., hits) were counted to generate a null model.

A Gaussian-shaped distribution of CNVs/5 Mb region was observed in the null model, but empirical data was enriched for observations at the extremities (Fig. 3B). Specifically, when empirical *P* values were calculated for each 5 Mb region, we found eighty-three 5 Mb regions (14.6%) where observed CNVs occurred more frequently than in the random model ("hotspots," *P* value <0.05) and fifty-six 5 Mb regions (9.9%) where empirical CNVs overlapped less frequently than in the null model ("cold spots," *P* value >0.99) (see Methods for *P* value determinations). For example, fourteen 5 Mb regions were hit at least 24 times by real CNVs, however this frequency (≥24 hits in a 5 Mb

region) occurred in only 0.5% of null model permutations. Importantly, no CNV-free region was observed in null model perturbations, but seven CNV-free cold spots were found in empirical data.

CNV hotspots and cold spots clustered in several semi-contiguous stretches of the genome (Fig. 3C). Eighty-three 5 Mb hotspots clustered into 47 distinct contiguous regions, whereas the 56 cold spots clustered into 22 distinct contiguous regions. Intriguingly, individual chromosomes also clustered as either hot or cold with respect to CNV presence or absence. For example, 9/83 (~11%) and 15/83 hotspots (~18%) clustered on chromosomes 18 and 5, respectively, whereas 12/56 cold spot regions (21%) clustered on chromosome 1. Thirteen highly aberrant neuronal genomes (containing ≥25 CNVs in empirical data) all had a CNV(s) that intersected hotspots, whereas only nine had CNVs intersecting cold spots. Similarly, of the 112 CNV neurons that contained between 1–5 CNVs, fifty-four had CNVs intersecting hotspots and only seven had CNVs intersecting cold spots. Overall, 163 neuronal genomes had a CNV(s) overlapping a hotspot, whereas only 50 CNV neurons overlapped cold spots.

One technical explanation for putative hotspots and cold spots is differential chromatin accessibility during WGA. For example, hotspots may simply be a consequence of limited chromatin accessibility leading to reduced genome amplification relative to coldspots. We assessed this possibility by counting open chromatin peaks (NeuN+ nuclei from DLPFC[56]) in each 5 Mb region and found the opposite association. Cold spots (3943 peaks in 56 regions, 70.4 ± 52.1 peaks/ region), which are consistently euploid and thus uniformly amplified, are associated (*P* < 0.0001 v. control) with fewer open chromatin peaks than control regions (54175 peaks in 404 regions, 134.1 ± 43.0 peaks/ region), and hotspots (12365 peaks in 83 regions, 149.0 ± 41.3 peaks/ region) are enriched (P < 0.006 v. control) in open chromatin (Supplementary Fig. 11C).

To extend these observation to other individuals and WGA approaches, we generated a random permutation model of the ref. 41 neuronal CNV atlas. Given that these 867 neurons represent a composite of 15 individuals ranging from <1 year-old to >90 years old, these data are unpowered to identify hotspots. However, cold spots may be

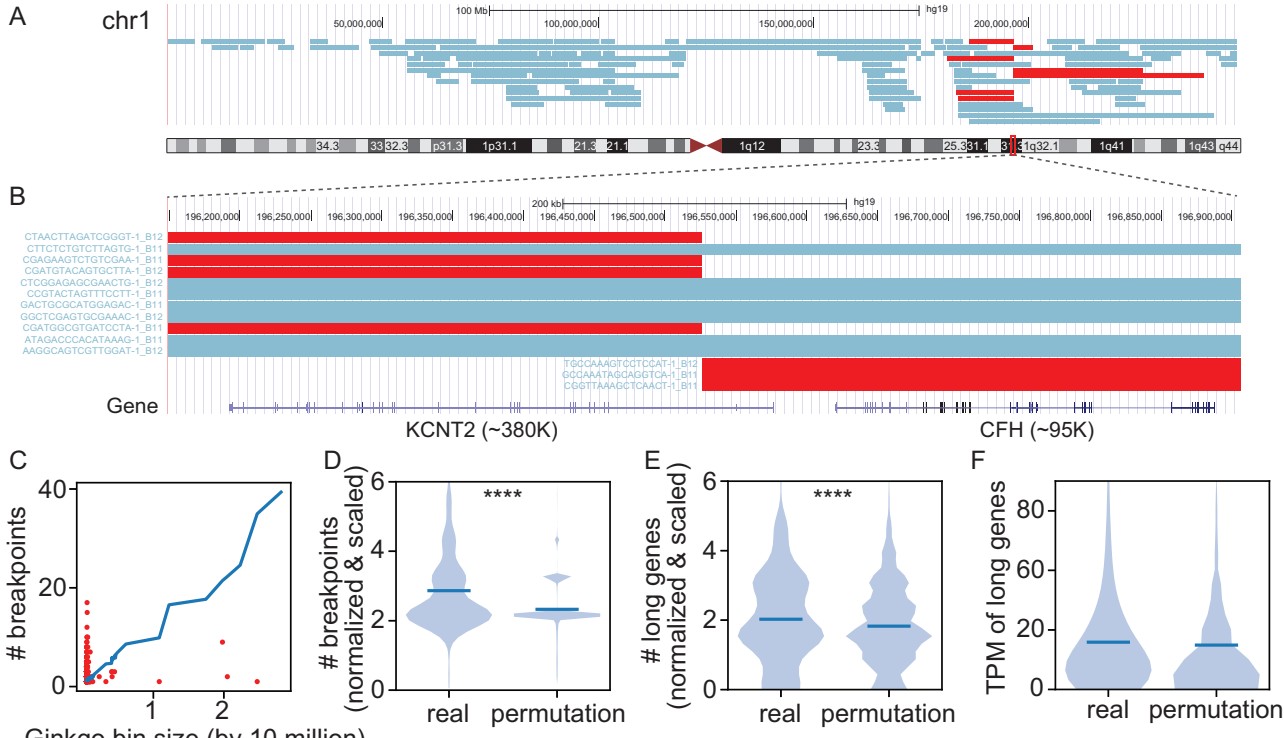

**Fig. 4 | Recurrent CNV breakpoints across multiple neurons. A** UCSC Genome Browser view of all CNVs detected on Chromosome 1 (47 neurons, rows). Seven neurons (red) contain CNVs that share a breakpoint region (CNVB). **B** Representative CNVB (red) on Chromosome 1 overlaps (±250 kb) two genes (lower panel). **C** Number of breakpoints identified in each Ginkgo bin (y-axis) relative to bin size (x-axis), shown for bins containing two or more CNVs (red) and averaged across all permutations in control set (blue line). **D**–**F** Violin plots show real and permuted datasets, normalized by bin size, when examined for **D** number of breakpoints, **E** number of long (>100k) genes (****$p < 0.0001$ for one-tailed $t$-test), and **F** transcripts per million bp (TPM) values of the longest gene in each bin.

conserved across individuals. As before, we generated a control dataset of randomly placed deletions and again observed that every 5 Mb region is overlapped by CNVs in the null model, whereas 58 5 Mb regions are not overlapped by real data (Supplementary Fig. 11D). Cold spots in the CNV atlas also cluster on few chromosomes and 40% of these overlap cold spots identified in this study (Supplementary Fig. 11E).

To examine if a neurobiological basis for a nonrandom distribution of CNVs in neuronal genomes may exist, we examined overlap between hotspot and cold spot regions, and 33 germline CNVs (fifty-six 5 Mb regions) that are associated with adverse neurodevelopmental phenotypes[57]. One-third (11/33) of these germline CNVs were in cold spots. By comparison, none (0/33) of the germline CNVs overlapped hotspots (Fig. 3C). The probability that a neuropathogenic germline CNV occurs in any 5 Mb genomic region by chance is approximately 33/567 (5.8%); however, empirical overlap was observed in 11/56 (19.6%) 5 Mb cold spot regions. Gene content further distinguished hotspots and cold spots from other control regions of the genome (Fig. 3D). Cold spots typically were gene-dense (64.7 ± 56.2 genes per 5 Mb region) and were not distributed uniformly when compared to control regions of the genome (Supplementary Fig. 11A). By comparison, hotspots typically were gene-sparse relative to cold spots (32.6 ± 15.2 genes per 5 Mb region).

## Recurrent regions of neuronal genome rearrangement

The observation that neuronal deletions cluster in genomic hotspots suggested that local genomic instability could, in principle, lead to recurrent mosaicism among neurons. Recurrent regions of genome rearrangement in cancer cells have led to the identification of drug targets (e.g., the Philadelphia chromosome and BCR-ABL[58]) and have been mechanistically associated with genomic fragile sites in long

genes[59,60]. To explore possible related mechanisms, we examined CNV start or end locations (i.e., breakpoints) that were shared amongst CNV neurons. Breakpoints are defined by one of the 5067 variably-sized Gingko bins that each include 500 kb of mappable sequence. Among these bins, 857 accounted for two or more CNV breakpoints (termed CNVBs) (Fig. 4A, B), many of which (220/851; ~26%) fell within previously identified hotspots.

We next sought to determine whether the number of bins containing more than two breakpoints was significantly different from a random CNV distribution (i.e., the control set of CNV permutations). Given variably-sized Gingko bins (Methods), we first assessed whether Ginkgo bin size impacted breakpoint frequency. While bin size scaled linearly with CNVB frequency in random permutations, this linear relationship was not observed with empirical CNVBs (Fig. 4C). When breakpoint counts are normalized by bin size, observed CNVBs cluster more frequently in common bins than random CNVBs (one-sided $t$-test, $P$ value: $2.08*e-134$), suggesting that CNVBs likely originate from a nonrandom process (Fig. 4D).

Empirical CNVBs were further assessed for properties that might suggest mechanisms of CNV formation. Recent studies have indicated that somatic CNV hotspots in non-cancer systems are localized around large (>500 kb) transcriptional units that form due to replication stress by a mechanism termed transcription-dependent double-fork failure[60,61]. These findings motivate the hypothesis[59,62,63] that longer genes incur additional DNA double strand breaks (DSBs) during transcripton which, in turn, lead to neuronal CNVs. Given that Ginkgo bins are imprecise relative to the sequence context around structural breakpoints[64], we restricted our analysis to larger genomic features. Gene content and gene expression levels were measured in CNVB regions relative to random CNV permutations. We observed a significant albeit modest enrichment of empirical CNVBs within long

genes (which we define as >100 kb as in ref. 65, one-sided *t*-test, *P* value: 1.32*e-5, 0.11-fold increase, Fig. 4E). However, gene expression levels in the 49-year-old postmortem brain were similar in CNVB regions relative to random permutations (Fig. 4F). Thus, neuronal CNVs could arise by related, but perhaps different, mechanisms associated with gene length.

Among 98 of the 226 CNV neurons, we observed 73 CNVs that shared both 3' and 5' CNVBs. These may be recurrent CNVs (CNVRs). Haplotype information was then used to determine if CNVRs support a clonal relationship among neurons. Briefly, we used phased allele ratios to compare whether CNVRs shared haplotypes by determining the median of the differences between the minimum and maximum log2 allele ratios observed in each SNP window within the CNVR across all cells where it was identified, reasoning that lower log2 allele ratio values would represent CNVRs on a shared haplotype (Methods, Supplementary Fig. 8A). These calculations resulted in two apparent distributions of both lower (32/73) and higher (41/73) delta log2 ratio values. The lowest delta log2 ratio cluster contained the two pairs of technical replicates (Fig. S5), indicating the veracity of our approach. The remaining CNVRs exhibited a delta median log2 ratio larger than 5, suggesting that these CNVs occurred on opposite haplotypes (Supplementary Fig. 8B). However, all CNV neurons harboring CNVRs had complex karyotypes with divergent CNV patterns across the genome (e.g., Fig. S13). These findings suggest that shared CNVs are not necessarily clonally-derived, but, instead, likely represent recurrent events (Supplementary Fig. 8C, D). Of note, similar CNVRs were observed in the analysis of cancer genomes and are referred to as "mirrored-subclonal" CNVs[47,66].

## Discussion

The genetic landscape of human neurons is a mosaic of the individual's germline genome; it is likely that every human neuron accumulates more than a thousand somatic variants over a person's lifetime[67–70]. Specific somatic mutations have been linked to overgrowth phenotypes in patients with hemimegalencephaly and focal cortical dysplasia[20,71–73]. Other studies report differential somatic mutation burden in subsets of patients with autism[17,23,62], schizophrenia[74], and neurodegenerative disease[75–77]. Mosaic Mb-scale CNVs alter the neurogenetic landscape in dramatic ways, yet it is unknown whether some genomic regions are more, or less, prone to CNV occurrence than other regions. The identification of CNV-prone or -resistant genomic loci, if they exist, could indicate mechanisms for somatic CNV formation, and, possibly, reveal a role for CNV neurons in brain function and disease.

Here we employed a droplet-based WGA approach to map CNVs in 2097 frontal cortical neurons from a single neurotypical individual. Technical barriers have limited previous studies to fewer than 100 neurons per individual and reported a total of 129 CNV neurons among 879 frontal cortical neurons examined from 15 individuals[41]. We developed SCOVAL to add veracity to read-depth-based CNV detection through an analysis of haplotype dropout. We showed high concordance between heterozygous deletions identified by read-depth and by phased LOH in single neuronal nuclei. In this sample, we found that 226/2097 (10.8%) of neurons harbor at least one Mb-scale CNV, and that 2% of CNV neurons were aneuploid. Moreover, we found that 65/226 CNV neurons contained highly aberrant karyotypes.

By combining haplotype and read-depth approaches, we have strong confidence that neuronal genomes contain large chromosomal segments that are not sampled using single cell sequencing approaches. This finding is consistent with previous reports that have examined a limited number of cells from neuronal and non-neuronal tissues using multiple technologies. Although we posit that the assayed sequence is missing because the corresponding segments have been deleted in vivo, unexpected technical or biological factors may yet contribute to the loss of signal. For example, neuronal preps exclude micronuclei[78]; however, the appreciable occurrence of micronuclei in neuronal tissue would still reflect an underlying alteration in genome content in the brain. Similarly, the lack of validated duplications in single-cell neuronal sequencing is striking. Ongoing development of new WGA approaches[79] and the application of long-read sequencing technologies to single-cell genomics[80] are poised to address these gaps in future studies. Furthermore, while some technologies for deriving long-range haplotype information are no longer commercially available (e.g., 10X Genomics linked-reads), the continued evolution and adoption of long-read sequencing for genome assembly and phasing[81,82] will provide a solid and improved foundation for additional single-cell studies using SCOVAL or similar strategies.

Our finding of a nonrandom distribution of 1861 deletions among 226 CNV neurons also allays concerns of random technical artifacts in neuronal CNV detection. Spurious WGA events, such as uneven genome amplification, are expected to occur randomly across the genome and are physically limited in size by the processivity of the polymerase (<20 kb). Multiple WGA approaches have been performed on single human neurons; all of these reported Mb-scale CNVs[38–41]. This technical concern was addressed previously[41] wherein a similar prevalence of CNV neurons was observed in two samples from the same individual (26-year-old), subjected to different WGA approaches. In Chronister et al., parameter optimization on synthetic datasets limited read-depth-based CNV detection to false positive rates <5%. Here, we provide additional lines of evidence that single-cell approaches for neuronal CNV detection are robust to technical artifacts. First, we showed that SCOVAL finds haplotype allele-level support for 76% of read-depth based deletion calls. Importantly, 99% of >10 Mb heterozygous deletions received orthogonal support via phased LOH. Second, when SCOVAL was applied to 2,097 neurons, the fraction of CNV neurons observed (10.8%) was concordant with the fraction (11.1%) identified using different chemistry on a smaller (99 neuron) sample from the same brain region. Perhaps most strikingly, we identified CNV hotspots and cold spots that were inconsistent with a random distribution of technical artifacts. Moreover, these data resolve disparate reports regarding aneuploid human neurons. Approaches that measured single (or few) chromosomes in each neuron suggested that >10% of neurons were aneuploid[6,7]. Extrapolations based on these data did not account for unmeasured chromosomes in the same neuron, implicitly assuming that every measured aneusomy represented a different aneuploid neuron. We identified 6 clearly aneuploid neurons, however, 52 neurons harbored deletions that covered >50% of atleast one chromosome and could reasonably be scored as aneuploid by traditional hybridization-based approaches. Taken together, these observations find a frequency of substantial chromosome loss (52/2095, 2.5%) in this individual that is consistent with other reports of neural aneuploidy[44,83].

In addition to finding a nonrandom distribution of CNVs among CNV neurons, we identified genomic hotspots that were impacted by neuronal CNVs more often than expected by chance; the same approach identified genomic cold spots. Further analysis of these regions found high gene density in cold spots (64.7 ± 56.2 genes per 5 Mb region), but a lower gene density (32.6 ± 15.2 genes per 5 Mb region) in hotspots. Complementary analysis identified 851 regions with 2 or more CNV breakpoints (i.e., CNVBs), and found that 220 of these refined previously defined 5 Mb hotspots to ±0.5 Mb. Hotspot CNVBs were enriched for long (>100Kb) genes, consistent with the paucity of genes found in these regions. In some cases, the functional consequences of the CNVs are also suggested by associations between long gene expression, neuronal development, and neuropathologies[84,85]. For example, we identified seven neurons with distinct CNVs sharing a breakpoint region within *KCNT2*, a long (~380 kb) gene that encodes an outward-rectifying potassium channel. *KCNT2* is important for neuron function and has been linked to several developmental pathologies[86–88] (Fig. 4B). *KCNT2* exhibited a TPM of

7.30, which falls within the expected range when considering the expression of all long genes in this tissue (mean TPM 9.56 ± 19.82).

Our study reveals that CNV neurons with highly aberrant karyotypes populate the neurotypical human frontal cortex. Although their impact on neural circuits and behavior remains unknown, cross-sectional studies indicate that CNV neurons are selectively vulnerable to aging-related loss[41]. The extent to which recurrent CNV sites are shared among individuals is not yet known; neither is it known if cold sites are refractory to CNV formation or are detrimental to neuronal survival during development. Nevertheless, we report candidate genomic regions that incur frequent neuronal gene rearrangement provides a rationale for tractable and scalable targeted single-cell sequencing. Many interesting questions follow from this study, including whether cold spots in neurotypical individuals are instead aberrant in individuals with neurological disease.

## Methods

### Sample and sequencing library preparation

The research in this project complies with all relevant ethical regulations. Postmortem human brain tissue was obtained at the time of autopsy via audiotaped witnessed informed consent from the legal next-of-kin allowing the use/sequencing of postmortem neurons/dural fibroblast tissue, through the Office of the Chief Medical Examiner of the State of Maryland, under the following two protocols: Maryland Department of Health IRB protocol #12–24 and the WCG protocol #20111080. We examined human neurons dissected from the dorsolateral prefrontal cortex (DLPFC) of a neurotypical individual (postmortem, 49-year-old male individual, LIBD: Br5154) used as the common reference brain in a previous study[37].

Neuronal nuclei (NeuN+) were purified from frozen tissue using a sucrose cushion and FANS (AF555conjugated anti-NeuN antibody, Millipore as in[41]. We then applied 10X Genomics Chromium Single Cell sequencing that ligated barcodes on the DNA in single cells within a Cell Bead Gel and the barcoded fragments are then pooled for library production, which can profile thousands of cells. We sequenced 2,125 neurons in two batches with a mean coverage of 0.114X (Fig. 1A). We further applied 10X Genomics Chromium Linked-Read sequencing to dural fibroblast tissue with very high sequencing coverage (52.7X) from the same individual to identify and phase germline SNPs by isolating and fragmenting long DNA segments into barcoded short reads that could be used to reconstruct underlying haplotypes using Long Ranger v2.2 (https://github.com/10XGenomics/longranger).

### Optimization of Ginkgo for single-cell CNV identification

The final CNV call set was generated using a combination of read-depth and phased loss-of-heterozygosity (LOH)-based validation. First, we processed read alignments from 2125 single cells using an adapted version of Ginkgo[45] to arrive at our unvalidated call set. The call set was then filtered via empirical *P* value selection using information pertaining to the loss of a particular haplotype, obtained by aligning sample reads to the (diploid) phased genome for this individual. The resultant calls were then filtered using a Bayesian classification model to arrive at the final CNV call set, which was further classified by CNV type (heterozygous deletions, homozygous deletions, and duplications) because the strength of support is different for these different CNVs, and the ensuing permutation testing (using heterozygous deletions alone) became more regularized. Only CNV calls in autosomes were included in the final CNV call set. We will now describe the generation pipeline, similar to ref. 41, in some detail.

### Setting CNV calling cutoffs in Ginkgo via the Gaussian Mixture Model.

Ginkgo was optimized by resetting default copy-number cutoffs that determine whether a segment detected by circular binary segmentation (CBS) will be called a CNV. To this end, we processed single-cell BAM files from 585 cells obtained from the five control individuals studied in[41] using the CBS implementation DNA-Copy (https://bioconductor.org/packages/release/bioc/html/DNAcopy.html). Aligned reads from each single cell were separately processed into 5067 autosomal bins across the hg19 human reference genome delineated by Ginkgo, which were then normalized to obtain an average copy number of two for the cell. We limited our analysis to autosomal bins to minimize false positives on monosomic allosomes in males. These individual bins were then grouped contiguously into segments based on similarity of their read coverage using DNACopy. We then fit a Gaussian Mixture Model (GMM) to the distribution of the median copy number of all segments from all cells using an "undoSD" of three, whereby two putative segments had to be more than three times the standard deviation in "intra-segment" copy number to be actually written as separate segments, and alpha = 0.01. From this fit, the two-tailed probability for the Gaussian curve centered at CN = 1 and the one at CN = 2 was calculated to be 1.63 (Supplementary Fig. 1B). This became the new copy-number cutoff for Ginkgo to call deletions. As seen in Supplementary Fig. 1B, there were not many candidate duplications to yield a proper fit, but the duplication cutoff was set at 2.43.

**Filtering to remove outlier bins via Tukey's rule.** Next, the raw bin CN data were filtered for the presence of uniform outlier bins across all cells (e.g., due to data-specific genomic regions uniformly subject to overamplification or underamplification, regions of poor mappability in the genome, etc.). The median of copy numbers of 2125 cells for each of the 5067 autosomal bins was first plotted. Tukey's rule was then applied to tag all bins whose median copy number exceeded Q3 + 1.5* IQR, or was below Q1-1.5*IQR, where the interquartile range IQR ≡ Q3-Q1 and Q1 and Q3 are the first and third quartiles, respectively, of all the median copy numbers. Three hundred and eight outlier bins were identified in addition to Ginkgo's original list containing 29 (Supplementary Fig. 1C). These bins were simply removed from the genome by Ginkgo prior to segment processing, while other bins (retaining their genomic coordinates) were merged. For reference, the genomic bin size used for benchmarking Ginkgo was 500 Kb. Thus, in this work, as in[41], we used Ginkgo settings pertaining to an approximate variable bin size of 500 Kb ("variable_500 kb_101_bowtie") and only considered large (>1 Mb) CNVs. Gingko reported a final mean bin size of 569 Kb, with bins ranging in size from 501 to 2812 Kb.

**Filtering of irregular cells.** For all cells, the mean absolute deviation (MAD) of bin copy numbers was calculated and fit to a Gaussian distribution. The mean (mu) and standard deviation (sigma) were 0.253 and 0.111, respectively. CNV calls from 19 cells (MAD > mu + 3* sigma) were removed before processing the data further (Supplementary Fig. 1A). The total number of reads for all remaining cells ranged uniformly from 580,809 to 8,983,573. However, one cell contained an inordinate proportion of reads (>80%) aligned to just one of the chromosomes and was removed. Further, eight cells that were not filtered by the above methods were manually curated from the dataset based on unlikely copy-number patterns, leaving a total of 2097 good neurons (see Supplementary Fig. 1D).

### Assessing the coverage-based single-cell CNV call set

To differentiate between bona fide CNVs and potential false positives due to coverage fluctuations, we leveraged the long-range haplotype information obtained from the 10X linked-read sequences generated from bulk analysis of matched dural fibroblast tissue. We made use of identified heterozygous SNPs (het-SNPs) and initially segmented the genome using phase blocks of heterozygous SNPs as identified by the linked-read data so that each segment would contain SNPs with consistent haplotype labeling. We then binned these segments further into windows of 20–100 SNPs based on empirical observations of SNP and read coverages. For each window in each cell, we then identified reads

that overlapped het-SNPs (herein termed "informative reads") and noted the allele present on the read. Notably, the coverage in each single cell resulted in a sparse number of informative reads per SNP window, typically resulting in 5–15 reads with specific allele information. Using the inferred haplotype of each overlapped het-SNP, we counted the number of reads present on each of the two haplotypes and calculated the absolute log2 ratio between the read counts if the total number of reads on each haplotype was larger than three. We used this log2 ratio to filter the CNV call set from the previous stage. First, we calculated the median log2 ratio of the windows within the CNV regions in the cells with those CNVs and the median log2 ratio of the windows within the CNV regions but in the cells without those CNVs as a background null model. From these data, we derived an empirical $p$ value for the observed log2 ratio in the sample with the CNV. We then collated the $p$ values for each individual CNV to derive a $p$ value distribution and selected a set of candidate CNVs with a $p$ value < 0.05.

Next, we randomly permuted 100 sets of "non-CNVs" size-matched to these candidate calls to build a GMM from the underlying median log2 ratios of each CNV/non-CNV region, with the assumption that the two distributions followed two distinct Gaussian distributions. Using the median absolute log2 ratios of the two datasets as the training data, we estimated the parameters of the Gaussians and predicted the posterior probability that the CNV belonged to the CNV distribution using a naive Bayesian classifier. Calls with posterior probability >0.99 were selected to process further.

As allele imbalance cannot support the homozygous deletions, we implemented a read-depth ratio measurement to add additional support to the calls. We calculated the read-depth ratio for each bin in every cell based on the bulk sequencing from the same tissue[28]. The read-depth ratio $RDR_{b,i}$ of bin b and cell i can be calculated as

$$RDR_{b,i} = \frac{C_{b,i}}{B_b} \frac{R_B}{R_i}, \qquad (1)$$

Where $C_{b,i}$ is the number of reads in bin b of cell i, $B_b$ is the number of the reads in bin b of bulk sequencing, $R_B$ is the total number of reads of bulk sequencing, and $R_i$ is the total number of reads of cell i. To distinguish between homozygous and heterozygous deletions, we applied a GMM on read-depth ratio to calculate the posterior probability for the homozygous deletions, and set the cutoff as >0.99 for posterior probability. The final call set for heterozygous deletions was obtained by adjudicating the above calls by requiring the CNV region to have an empirical median log2 ratio $p$ value (as described above) to be less than 0.01 (thus ensuring that only calls in regions showing the highest relative allelic preference were selected).

### Germline CNV assessment

To determine whether any potential germline CNVs were included in our analysis, we analyzed the 10X linked-read sequences using both Manta and Long Ranger (https://github.com/10XGenomics/longranger) using default settings and compared the detected CNVs with our somatic CNVs using a 50% reciprocal overlap criteria. We further examined the minor allele frequencies of heterozygous SNPs across all cells within the boundaries of detected somatic CNVs with the expectation that germline deletions would have a consistent deviation from 0.50 frequency if present.

### Benchmarking CNV detection

We applied CHISEL[47] to our single-cell sequencing data with its default parameters (max balanced ploidy = 4); however, it reported unrealistic results. Only 8.16% of all 5MB windows were reported as normal diploid regions with haplotype copy number "1|1", with most windows (77.83%) indicating the max balanced ploidy with haplotype copy number "2|2". We adjusted the max balanced ploidy setting to 2, resulting in 98.15%

of the windows now indicated as normal diploid regions. We combined neighboring CNV windows within the same cell to calculate the overlap percentage with our final call set.

### Clonal cells and recurrent CNVs

To detect the clonal structure of neurons based on CNVs, we designed a very conservative method to identify clonal events. We first found all the CNVs that shared the same start and end breakpoints, then we marked these loci as CNVR. With the haplotype information, we could identify whether these loci were clonal events or the recurrent events that existed on the different haplotypes. For each bin covered by the CNVR, we took the maximum log2 ratio and minimum log2 ratio of the cells with the CNVR and calculated the delta log2 ratio using maximum minus minimum. Next, we calculated the median delta log2 ratio across the bins for each CNVR and observed two distinct distributions, one representing potential clonal events (low delta log2 ratio; CNVs are on the same haplotype) and the other indicating likely independent events (high delta log2 ratio; CNVs are on the different haplotypes).

### Characterizing CNV neurons

**Neuronal distribution of CNVs.** The raw distribution of the number of CNVs per neuron is shown as a histogram (Fig. 2A) on a log scale, along with a null model based on a uniform random distribution of all CNVs in the final call set across all good neurons. Thus, a Poisson curve with mean = (# final CNVs)/(# good cells), scaled up by the total number of good neurons, was superimposed on the first plot to assess whether the final call set contained more CNV-rich neurons than expected by a uniform distribution.

**Hierarchical clustering and complex karyotypes.** The 2097 good neurons were ordered based on the number of total base pairs affected by heterozygous deletions in descending order. A heat map of all cells was generated showing the percentage of base pairs affected by heterozygous deletions in each autosome (see Fig. 2C), Neurons were sorted and numbered in reverse order of % base pairs affected. Those cells affected by more than 5% were termed complex neurons and numbered 1–65 in our call set. All good neurons were clustered using hierarchical clustering using each autosome as an independent dimension and the percentage of base pairs affected as the distance measure. Thus, cells with chromosomes that were similarly affected by heterozygous deletions clustered together (Fig. 2D). Some cells with possibly multiple recurrent events were identified (Fig. 2E), and some seemingly clonal cells were analyzed to be technical replicates.

**Identifying CNV hotspots and cold spots via permutation testing.** The final heterozygous deletion call set was "shuffled" using bedtools[89] to arrive at 10,000 unique synthetic permutations (Fig. 3A). In each permutation, CNVs in each cell were permuted uniformly at random in the autosomes while prohibiting collision ("noOverlapping" option) and then assembled together. The process was repeated 10,000 times without genomic constraints, as unmappable regions were a priori removed (refer to subsection Optimization of Ginkgo for single-cell CNV identification), and calls "straddling" such regions commonly occurred in the final call set (Supplementary Data 2).

Each autosome was divided into contiguous 5 Mb regions (the remaining smaller tails of chromosomes were not considered). The number of unique hits (defined as simple overlap) of each region with synthetic CNVs from all 10,000 permutations was recorded, resulting in a CNV distribution profile for the synthetic data. For each 5 Mb region, a $P$ value was assessed for the number of CNV hits in real data among the 10,000 hit values in the region's synthetic CNV profile. For our purposes, we define $P$ value to be the fraction of simulated instances that were at least as high as the real number of CNV hits to the 5 Mb region. Given that CNV hits are discrete-valued, and we are using the same definition of $P$ value for cold spots and hotspots, we

impose a more stringent cutoff for cold spots to account for the inherent liberal treatment of data values on the lower extreme (which may lead to an overabundance of cold spots). Regions with a *P* value <0.05 (i.e., where hits were among the top 5% of synthetic hit-values for that region) were termed "hotspots" and those with *P* value >0.99 were termed "cold spots." Regional significance (defined as 1 - *P* value) was plotted against the autosomal genome on the x-axis (Supplementary Fig. 9). The distribution of the raw number of CNV hits in 5 Mb regions is shown in Fig. 3B. Cold spots were screened for aberrant genomic blocks that might hamper CNV calling or regions a priori neglected. To this end, cold spot regions were coordinate-merged (via "bedtools merge") and compared to all a priori removed bad bins as well as blacklisted regions[90] by means of a relative permutation analysis. A merged cold spot that overlapped more with blacklisted regions and bad bins as appropriately compared to 1000 randomly selected non-cold spot intervals was removed from the list of final cold spots (the cutoff chosen was *p* > 0.05) (Supplementary Fig. 10A). Each merged cold spot was mapped to 1000 randomly selected regions other than existing cold spots, and its overlap with bases contained in bad bins and blacklisted regions, respectively, were calculated in each instance in order to assign it a *p* value. For additional relevant detail, some genomic heat maps of the copy number of CNV neurons are shown in Supplementary Fig. 10C, D along with merged cold spots and bad bins. For rigor, cold spots were analyzed for the presence of deduplicated germline structural variants from 1000 individuals from FusorSV[91], the cold spots had a larger SV coverage (11.4) than the unremarkable regions (7.25), further supporting that CNVs are callable in these regions.

Hotspots and cold spots are shown throughout the genome in a Circos[92] plot along with 33 regions of the genome where germline CNVs are associated with neurodevelopmental phenotypes[57] to assess any possible correlation between the two (Fig. 3C). The distribution of the number of genes in 5 Mb regions was also plotted for hotspots, cold spots and unremarkable regions as control (Fig. 3D). Similar distributions were plotted (with assigned *p* values) for long genes and different expression levels (Supplementary Fig. 11).

In a complementary assessment, the above permutation analysis was repeated for genes instead of 5 Mb genomic regions. To profile gene expression, histograms of *p* values for genes were shown for different gene expression categories (Supplementary Fig. 12) to assess/confirm the general prevalence of hotspots and cold spots in each expression category.

**Recurrent CNV breakpoint analysis.** To assess the impact of different Ginkgo bin sizes on the CNV breakpoint distribution, we used the previously described 10 K permuted CNV sets to determine the relationship between the number of breakpoints and Ginkgo bin size. We calculated the mean of the number of breakpoints from all permuted CNVs and compared this to the size of the Ginkgo bin in which they fell. We then normalized the number of breakpoints by the Ginkgo bin size and compared this normalized number of observed breakpoints within CNVB regions with those in permuted regions using a one-sided *t*-test with the alternative hypothesis that observed > permuted. We then calculated the normalized number of long genes (>100 K) overlapped with CNVB bins and compared against the permuted regions using the same strategy. The gene expression analysis was conducted by calculating the transcript per million (TPM) values for the longest gene observed in each of the CNVB and permuted regions and assessing whether they were significantly different using a one-tailed *t*-test.

**Reporting summary**
Further information on research design is available in the Nature Portfolio Reporting Summary linked to this article.

## Data availability
The single cell and linked-read sequencing data and call sets generated in this study have been deposited in the National Institutes of Mental Health (NIMH) Data Archive under Study ID 1680 (https://doi.org/10.15154/1527774). These can be accessed as part of the NIMH Data Archive permission groups (https://nda.nih.gov/user/dashboard/data_permissions.html). Data obtained from ref. 41 is available through Synapse (https://www.synapse.org/#!Synapse:syn16803262) and through the NIMH data archive (https://nda.nih.gov/edit_collection.html?id=2963 and https://nda.nih.gov/edit_collection.html?id=2458). To promote the responsible use of shared data, all institutions and investigators seeking access must commit to comply with NDA policies and procedures by signing a Data Use Certification. Initial single-cell Ginkgo calls, subsequent SCOVAL assessments, and CHISEL outputs are included in Supplementary Data Files 1–3, respectively.

## Code availability
The workflow to generate the final call set is available at https://github.com/mills-lab/Scoval.

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

## Acknowledgements

We thank Drs. TE Wilson and FH Gage for essential insight and helpful critique throughout the study, and ML Gage for editorial assistance. We also thank M Wolpert and M Haakenson for their technical assistance. Members of the BSMN consortium are listed at the end of the article. G. Senthil, M Gitik, and T Lehner organized the BSMN consortium. The genome analysis and technology, and flow cytometry cores at the University of Virginia School of Medicine assisted with sample preparation. This work was supported by NIMH funding to J.V.M., J.M.K., and R.E.M. (U01 MH106892), J.V.M. (U01 MH106892 supplement), D.R.W. (U01 MH106893), and M.J.M. [FH Gage (PI), U01 MH106882]. I.E.B. is supported by (FONDECYT Regular 1191737 Agencia Nacional de Investigación y Desarrollo de Chile).

## Author contributions

M.J.M., R.E.M., J.V.M., J.M.K., and D.R.W. designed the study. S.B.E., I.E.B., J.H.S., J.M.K., and M.J.M. generated sequencing data. C.S., K.K., B.K., J.M.K., R.E.M., and M.J.M. performed data analysis. C.S., K.K., J.V.M., J.M.K., R.E.M., and M.J.M. wrote the manuscript.

## Competing interests

J.V.M. is an inventor on patent US6150160, is a paid consultant for Gilead Sciences, serves on the scientific advisory board of Tessera Therapeutics Inc. (where he is paid as a consultant, and has equity options), has licensed reagents to Merck Pharmaceutical, and recently served on the American Society of Human Genetics Board of Directors. The remaining authors declare no competing interests.

## Additional information

# Brain Somatic Mosaicism Network

Joseph G. Gleeson[9], Martin W. Breuss[9], Xiaoxu Yang[9], Danny Antaki[9], Changuk Chung[9], Dan Averbuj[9], Laurel L. Ball[9], Subhojit Roy[9], Daniel Weinberger[2], Andrew Jaffe[2], Apua Paquola[2], Jennifer Erwin[2], Joo Heon Shin[2], Michael J. McConnell[2]✉, Richard Straub[2], Rujuta Narurkar[2], Gary Mathern[10], Christopher A. Walsh[11], Alice Lee[11], August Yue Huang[11], Alissa D'Gama[11], Caroline Dias[11], Eduardo Maury[11], Javier Ganz[11], Michael Lodato[11], Michael Miller[11], Pengpeng Li[11], Rachel Rodin[11], Rebeca Borges-Monroy[11], Robert Hill[11], Sara Bizzotto[11], Sattar Khoshkhoo[11], Sonia Kim[11], Zinan Zhou[11], Peter J. Park[12], Alison Barton[12], Alon Galor[12], Chong Chu[12], Craig Bohrson[12], Doga Gulhan[12], Elaine Lim[12], Euncheon Lim[12], Giorgio Melloni[12], Isidro Cortes[12], Jake Lee[12], Joe Luquette[12], Lixing Yang[12], Maxwell Sherman[12], Michael Coulter[12], Minseok Kwon[12], Semin Lee[12], Soo Lee[12], Vinary Viswanadham[12], Yanmei Dou[12], Andrew J. Chess[13], Attila Jones[13], Chaggai Rosenbluh[13], Schahram Akbarian[13], Ben Langmead[14], Jeremy Thorpe[14], Sean Cho[14], Alexej Abyzov[15], Taejeong Bae[15], Yeongjun Jang[15], Yifan Wang[15], Cindy Molitor[16], Mette Peters[16], Fred H. Gage[17], Meiyan Wang[17], Patrick Reed[17], Sara Linker[17], Alexander Urban[18], Bo Zhou[18], Reenal Pattni[18], Xiaowei Zhu[18], Aitor Serres Amero[19], David Juan[19], Inna Povolotskaya[19], Irene Lobon[19], Manuel Solis Moruno[19], Raquel Garcia Perez[19], Tomas Marques-Bonet[19], Eduardo Soriano[20], John V. Moran[3], Chen Sun[1,23], Diane A. Flasch[3], Trenton J. Frisbie[3], Huira C. Kopera[3], Jeffrey M. Kidd[1,3], John B. Moldovan[3], Kenneth Y. Kwan[3], Ryan E. Mills[1], Sarah B. Emery[3], Weichen Zhou[1], Xuefang Zhao[1], Aakrosh Ratan[21], Flora M. Vaccarino[22], Adriana Cherskov[22], Alexandre Jourdon[22], Liana Fasching[22], Nenad Sestan[22], Sirisha Pochareddy[22] & Soraya Scuder[22]

[9]Department of Neurosciences, University of California, San Diego, La Jolla, CA, USA. [10]University of California, Los Angeles, Los Angeles, CA, USA. [11]Boston Children's Hospital, Boston, MA, USA. [12]Harvard University, Boston, MA, USA. [13]Icahn School of Medicine at Mount Sinai, New York, NY, USA. [14]Kennedy Krieger Institute, Baltimore, MD, USA. [15]Department of Quantitative Health Sciences, Mayo Clinic, Rochester, MN, USA. [16]Sage Bionetworks, Seattle, WA, USA. [17]Salk Institute for Biological Studies, La Jolla, CA, USA. [18]Stanford University, Stanford, CA, USA. [19]Universitat Pompeu Fabra, Barcelona, Spain. [20]University of Barcelona, Barcelona, Spain. [21]University of Virginia, Charlottesville, VA, USA. [22]Yale University, New Haven, CT, USA.

