## [Peer Review File · Nature Communications]

Mapping Recurrent Mosaic Copy Number Variation in Human NeuronsReviewer #1 (Remarks to the Author):

Sun, Kathuria et al. use 10X Chromium scWGS and an allele-based validation approach (SCOVAL) to survey the copy number variant (CNV) landscapes of 2125 cortical neurons from one neurotypical individual. At ~0.1X coverage per neuron, the authors report approximately 10% of neurons harboured at least one megabase sized CNV. The vast majority of the detected CNVs were heterozygous deletions. A subset of the CNV breakpoints were found in multiple neurons but, based on genome-wide CNV profiles, these were inferred as being recurrent CNVs, as opposed to clonal CNVs. Finally, the authors test whether the genome-wide distribution of CNVs is non-random, finding a significant enrichment in long genes (>100kb).

I do not have any major criticisms to offer. The use of phased high-confidence SNPs (with SCOVAL) to assess CNVs is an appealing approach, made more robust by focusing on megabase scale heterozygous deletions. The writing and figures are clear, and I would compliment the authors on the thoughtful Discussion section, outlining the pros and cons of their study, and how it fits in the CNV literature. It is good to see that the estimates of CNV frequency are increasingly consistent across studies. There is still work to be done of course, but being able to accurately detect the most common type of CNVs in thousands of neurons is a valuable advance for the field.

Minor points:

- 1) The authors should state the absolute enrichment they observe for CNVs (or CNV breakpoints) in long genes. It looks relatively modest, even if significant by permutation test. Also, did this analysis check for enrichment in annotated A/B compartments? Presumably there are hot spots in the B compartment?
- 2) The amplification approach is quite an important part of the study, and it would be useful to readers if this was described in more detail in the Methods.
- 3) Could a brief explanation be provided to readers as to why only autosomal CNVs (line 478) were retained?
- 4) Please cite PMID:31230816 on line 62. The "small subset" part is debatable given these retrotransposition events (≤ 1 in pan-neuronal sorted populations) could be more frequent than CNVs. Could simply say they happen ... will leave this up to the authors.
- 5) The Discussion does a nice job describing how difficult it is to validate heterozygous deletions in single neurons. The authors may wish to add more specifics about what could be done in this space in the future. For instance, haplotype-resolved long read sequencing could presumably reconstruct breakpoints at higher resolution, assuming the deletion creates a new junction to recover.

Geoff Faulkner (University of Queensland)

Reviewer #2 (Remarks to the Author):

Sun and colleagues develop allele-based validation approach SCOVAL to validate somatic CNV's within neurons; identified 226 CMNV neurons; and found the CNV location was non-random. Several major concerns dampen enthusiasm. The single largest issue is the lack of validation using orthogonal methods on gold standard data approaches, such as spectral karyotyping. The second largest concern is that the impact and significance of core findings is low.

1. Loss of heterozygosity is the approach of validation, but they are using the same data as their discovery approach. Many algorithms use direct or indirect allelic imbalance, and it's not clear whether this represents an independent validation. At best, its evidence is consistent with the initial

observation.

2. Power, sequencing data, etc - or the ability to resolve CNV events is non-random, and thus it's not clear that the finding of non-randomness of CNV location is not simply an artifact. Again, an experimentally distinct validation would be better.

3. The method isn't wholly novel, and is overly dependent on an assay type (10X phased reads) that is no longer commercially available and has been depreciated.

4. There are expected hot-spots, and this has been observed in karyotype data in the 80s and 90s. The authors should really consider reviewing and looking into our understanding of somatic copy number from spectral karyotyping methods.

5. With above, there is no independent validation and it's not clear if the results are not just an artifact.

Reviewer #3 (Remarks to the Author):

Title: Mapping the complex genetic landscape of human neurons

In this study, the authors employed a droplet-based WGA approach with an allele-based validation to identify 226 CNV neurons out of 2,125 cortical neurons from a neurotypical human brain.

Furthermore, they found that CNV location appears to be non-random. Instead, there were CNV hotspots and cold spots in the genome. Recurrent CNV regions contained fewer, but longer genes. The issue of developing a method which accurately identifies single-cell CNVs is very important in the field of somatic mosaicism. In this regard, this study is important and interesting. However, this reviewer is not fully convinced by the result and conclusion of this study due to a couple of reasons. Firstly, as the authors pointed out, WGA events can result in a lot of artifactual CNVs. Although they rigorously applied SCOVAL for the identification of true CNVs in a single cell, their result of CNV hotspots and cold spots should be validated in the different WGA methods such as PTA. Secondly, they sequenced neurons from one person and generalized their finding to human neurons. Again, their result should be validated in other individuals. Taken together, in order to agree with the author's conclusion, this reviewer would like to suggest the use of another WGA method (e.g PTA) in a different cohort.

Other comments;

1. Figure 2 could be more informative. Addition of labelling the specific neurons (specifically the monosomic neurons) in Figure 2 A/B would aid in understanding.

2. Clarification of Figure 2C - "monosomic neurons harboured deletions affecting >40% of other chromosomes" - but I cannot see this in the figure.

3. In figure 2D, since the CNVs are apparently not clonal, is the tree informative?

4. Better explanation and experimentation on how they can combat CNV artifacts - since the conclusions drawn rely on this.

5. Can they be 100% certain that germline CNVs were not included? Germline CNVs should be detected in many of single neurons.

6. The authors should mention how they stain Neun and the FANS plot is needed in the supple figures.

RE: NCOMMS-23-09510A
RE: Response to Reviewer's

December 24, 2023

Chen, *et al.* Mapping the Complex Genetic Landscape of Human Neurons

Below we have interspersed responses with the original reviewer critiques. Our responses are in red text and *new manuscript text* is in blue italics.

REVIEWER COMMENTS

Reviewer #1 (Remarks to the Author):

Sun, Kathuria et al. use 10X Chromium scWGS and an allele-based validation approach (SCOVAL) to survey the copy number variant (CNV) landscapes of 2125 cortical neurons from one neurotypical individual. At ~0.1X coverage per neuron, the authors report approximately 10% of neurons harboured at least one megabase sized CNV. The vast majority of the detected CNVs were heterozygous deletions. A subset of the CNV breakpoints were found in multiple neurons but, based on genome-wide CNV profiles, these were inferred as being recurrent CNVs, as opposed to clonal CNVs. Finally, the authors test whether the genome-wide distribution of CNVs is non-random, finding a significant enrichment in long genes (>100kb).

I do not have any major criticisms to offer. The use of phased high-confidence SNPs (with SCOVAL) to assess CNVs is an appealing approach, made more robust by focusing on megabase scale heterozygous deletions. The writing and figures are clear, and I would compliment the authors on the thoughtful Discussion section, outlining the pros and cons of their study, and how it fits in the CNV literature. It is good to see that the estimates of CNV frequency are increasingly consistent across studies. There is still work to be done of course, but being able to accurately detect the most common type of CNVs in thousands of neurons is a valuable advance for the field.

We thank Dr. Faulkner for his consideration of our work, its impact, and our discussion of existing caveats. We agree that there are still hypotheses to be tested with regards to genome stability and potential functional impact of these somatic events, and hope that our improved analysis approaches and results will continue to motivate additional studies in this area.

Minor points:

1) The authors should state the absolute enrichment they observe for CNVs (or CNV breakpoints) in long genes. It looks relatively modest, even if significant by permutation test. Also, did this analysis check for enrichment in annotated A/B compartments? Presumably there are hot spots in the B compartment?

We agree that although statistically significant, the overall enrichment of CNV breakpoints in longer genes is indeed quite modest (0.11-fold difference). We expect this is likely limited by the low resolution of breakpoint positions (+/- ~500Kb) as well as the number of detected CNV neurons which, while larger than previous studies, is still only a small representation of cortical tissue. The future development of new technologies, for example single cell WGA using long reads, should enable a more precise detection of such breakpoints and will allow a refinement of our results. This paragraph has been re-written as follows:

Empirical CNVBs were further assessed for properties that might suggest mechanisms of CNV formation. Recent studies have indicated that somatic CNV hotspots in non-cancer systems are localized around large (>500kb) transcriptional units that form due to replication stress by a mechanism termed transcription-dependent double-fork failure^{60,61}. These findings motivate the hypothesis^{59,62,63} that longer genes incur additional DNA double strand breaks (DSBs) during transcription which, in turn, lead to neuronal CNVs. Given that Ginkgo bins are imprecise relative to the sequence context around structural breakpoints⁶⁴, we restricted our analysis to larger genomic features. Gene content and gene expression levels were measured in CNVB regions relative to random CNV permutations. We observed a significant albeit modest enrichment of empirical CNVBs within long genes (which we define as >100kb as in Zylka, et al.⁶⁵, one-sided t-test, P-value: 1.32×10^{-5} , 0.11 fold increase, **Fig. 4E**). However, gene expression levels in the 49 year-old post-mortem brain were similar in CNVB regions relative to random permutations (**Fig. 4F**). Thus, neuronal CNVs could arise by related, but perhaps different, mechanisms associated with gene length.

We had not previously considered examining whether CNV hotspots or cold spots fell within open or closed chromatin, but this is an excellent suggestion. We have now explored this using the Fullard *et al.* data of NeuN+ nuclei from human DLPFC. ATAC peaks were counted in each of the 5Mb regions used to identify hotspots and cold spots. We find that, relative to control regions, cold spots have significantly fewer open chromatin peaks and hotspots have significantly more. This analysis and how it relates to potential WGA artifacts is discussed in results and presented in Supplementary Figure 11C.

One technical explanation for putative hotspots and cold spots is differential chromatin accessibility during WGA. For example, hotspots may simply be a consequence of limited chromatin accessibility leading to reduced genome amplification relative to cold spots. We assessed this possibility by counting open chromatin peaks (NeuN+ nuclei from DLPFC⁵⁶) in each 5Mb region and found the opposite association. Cold spots (3943 peaks in 56 regions, 70.4 ± 52.1 peaks/region), which are consistently euploid and thus uniformly amplified, are associated ($P < 0.0001$ v. control) with fewer open chromatin peaks than control regions (54175 peaks in 404 regions, 134.1 ± 43.0 peaks/region), and hot spots (12365 peaks in 83 regions, 149.0 ± 41.3 peaks/region) are enriched ($P < 0.006$ v. control) in open chromatin (**Supplementary Fig. 11C**).

2) The amplification approach is quite an important part of the study, and it would be useful to readers if this was described in more detail in the Methods.

We thank the reviewer for this suggestion and now describe the 10X WGA approach in more detail in the first paragraph of Results.

Here, we employ a fourth WGA approach (10X Genomics Single Cell CNV) that uses droplet-based microfluidics to enable analysis of hundreds to thousands of single nuclei from a single sample. In this approach, WGA is performed on thousands of nuclei, each individually encapsulated in a hydrogel. Hydrogel beads retain amplified genomic DNA, and are then microfluidically paired with barcodes, leading to a library pool containing hundreds to thousands of single nuclei.

3) Could a brief explanation be provided to readers as to why only autosomal CNVs (line 478) were retained?
“d. Only CNV calls in autosomes were included in the final CNV call set.”

We have added this sentence to the third paragraph of Methods.

We limited our analysis to autosomal bins to minimize false positives on monosomic sex chromosomes in males.

4) Please cite PMID:31230816 on line 62. The "small subset" part is debatable given these retrotransposition events (≤ 1 in pan-neuronal sorted populations) could be more frequent than CNVs. Could simply say they happen ... will leave this up to the authors.

LINE-1 mobile elements retrotranspose during neurogenesis and contribute to brain somatic mosaicism in a small subset of neurons 29, 30, 31, 32, 33 62.

We thank the reviewer for this suggestion and have now incorporated this reference into our manuscript and we removed the word 'small' from this sentence.

Long Interspersed Element-1 (LINE-1) mobile elements retrotranspose during neurogenesis and contribute to brain somatic mosaicism in a subset of neurons³⁰⁻³⁵.

5) The Discussion does a nice job describing how difficult it is to validate heterozygous deletions in single neurons. The authors may wish to add more specifics about what could be done in this space in the future. For instance, haplotype-resolved long read sequencing could presumably reconstruct breakpoints at higher resolution, assuming the deletion creates a new junction to recover.

We thank the reviewer for this comment, and have now added some additional prospective comments in the third paragraph in Discussion.

Ongoing development of new WGA approaches⁷⁹ and the application of long read sequencing technologies to single cell genomics⁸⁰ are poised to address these gaps in future studies. Furthermore, while some technologies for deriving long-range haplotype information are no longer commercially available (e.g., 10X Genomics linked-reads), the continued evolution and adoption of long read sequencing for genome assembly and phasing^{81,82} will provide a solid and improved foundation for additional single-cell studies using SCOVAL or similar strategies.

Geoff Faulkner (University of Queensland)

We thank Dr. Faulkner for his evaluation of our work and the helpful comments he provided to improve the clarity and scope of our manuscript.

Reviewer #2 (Remarks to the Author):

Sun and colleagues develop allele-based validation approach SCOVAL to validate somatic CNV's within neurons; identified 226 CMNV neurons; and found the CNV location was non-random. Several major concerns dampen enthusiasm. The single largest issue is the lack of validation using orthogonal methods on gold standard data approaches, such as spectral karyotyping. The second largest concern is that the impact and significance of core findings is low.

We thank the reviewer for their critique of our work. Whole genome amplification was first applied to single neurons in 2013. This initial study analyzed a total of 110 neurons from 3 individuals and identified 45 CNV neurons (41%). This study was contemporaneous with others (e.g., Navin, et al) that applied the same approach to cancer cells where clonal CNVs were abundant. As the reviewer suggests, clonal CNVs in these studies, that were known from cytogenetic characterization of the cell line, provided gold standard validation of whole genome amplification and short read sequencing.

1. Loss of heterozygosity is the approach of validation, but they are using the same data as their discovery approach. Many algorithms use direct or indirect allelic imbalance, and it's not clear whether this represents an independent validation. At best, its evidence is consistent with the initial observation.

We acknowledge that the use of allelic information does not represent a truly independent validation of our read depth ascertained CNVs, however it does provide orthogonal support that would be expected to be mostly, though not entirely, uncoupled from the overall depth of coverage. We also note that we are not using loss of heterozygosity in the traditional sense; that is, we do not have the per-cell sequencing coverage to expect to see more than 1 allele at each position and thus cannot ascertain whether another allele should be present but isn't. Rather, we instead are using long-range phasing information from independent data to identify loss of heterozygosity of a specific haplotype over a large span of the genome. While this does not completely rule out technical artifacts of uneven whole genome amplification that might lead to false positive calls, we feel the consistency of our results with prior studies using several different WGA and microarray methodologies as shown below indicates that such technology-specific errors would likely be rare.

2. Power, sequencing data, etc - or the ability to resolve CNV events is non-random, and thus it's not clear that the finding of non-randomness of CNV location is not simply an artifact. Again, an experimentally distinct validation would be better.

5. With above, there is no independent validation and it's not clear if the results are not just an artifact.

Reviewer concerns 2 and 5 are related and addressed together here. We appreciate the reviewer's concerns regarding the potential artifacts in our results. Given that individual cells are destroyed during the whole genome amplification process, it is impossible to go back and directly validate single cell results outside of interrogating the WGA library, which may intrinsically harbor its own biases.

Instead, we re-analyzed previous studies that identified CNV neurons using distinct technologies, as the reviewer notes, to determine whether our observations are consistent. We have now extended our observations to other

individuals and WGA approaches by using data from the Chronister, et al., which compiled a database of 867 neurons from which 80 harbored CNVs. These data were generated by four laboratories using three different WGA approaches, and between 20 and 120 NeuN+ nuclei isolated from post-mortem and acutely resected frontal cortex from 15 individuals. We used these data to derive a null model of expected background CNV prevalence. Given the broad age range and small per individual sample size, these data are under-powered to detect hotspots. However, our null model finds no 5Mb region with 0 CNV overlaps; whereas 58 5Mb regions are not overlapped by real data. The Chronister-derived cold spots also cluster on few chromosomes and 40% of these overlap cold spots found in the common control brain.

SCOVAL produced a final deletion CNV set (Supplementary Table 2) comprising 1,957 somatic CNV calls (13.95 Mb +/- 17.47 Mb) among 226 CNV neurons (~11%). These represent 76.3% of the initial 2,564 read depth predictions. Notably, CNV neuron prevalence (226/2097 neurons) using droplet-based WGA (10X) and SCOVAL is in good agreement with previous read-depth based CNV detection using an alternative WGA approach (Picoplex) from this individual (~11%; 11 of 99 neurons)⁴¹.

To extend this observation to other individuals and WGA approaches, we generated a random permutation model of the Chronister, et al.⁴¹ neuronal CNV atlas. These 867 neurons represent a composite of 15 individuals ranging from <1 year-old to >90 years-old and hotspots may vary among individuals of different ages. However, because CNV neuron prevalence declines with age cold spots may be conserved across individuals. As before, we generated a control dataset of randomly placed deletions and again observed that every 5 Mb region is overlapped by CNVs in the null model, whereas 58 5Mb regions are not overlapped by real data (Supplementary Fig. 11D). Cold spots in the CNV atlas also cluster on few chromosomes and 40% of these overlap cold spots identified in this study (Supplementary Fig. 11E).

3. The method isn't wholly novel, and is overly dependent on an assay type (10X phased reads) that is no longer commercially available and has been depreciated.

We agree with the reviewer that our approach is not completely novel in terms of overall strategy. However, we note that almost all other similar approaches were developed for application to cancer tissue, with the expectation of a higher degree of clonality to enable somatic CNV discovery. We have tailored our approach to specifically identify somatic CNVs in single neurons, and we show in our manuscript that methods designed for cancer are not necessarily applicable for lower frequency variants lacking clonal support (see CHISEL comparison in “Determining the genetic architecture of individual neurons”).

We appreciate the reviewer's knowledge of current genomics technologies and recognize that assays that were available at the start of our project (i.e. 10X Genomics linked-read sequencing) are no longer commercially available. Our approach is not dependent on 10X linked-reads per se, but rather relies on having accurate long-range phased haplotypes which can be obtained from many different available assays including other linked-read technologies (e.g. Tell-Seq), long-read whole genome sequencing, or even imputation. As such, we do not see this as a limitation to our approach. We have updated the text in the Discussion section to note these alternatives.

Furthermore, while some technologies for deriving long-range haplotype information are no longer commercially available (e.g., 10X Genomics linked-reads), the continued evolution and adoption of long read sequencing for genome assembly and phasing will provide a solid and improved foundation for additional single-cell studies using SCOVAL or similar strategies.

4. There are expected hot-spots, and this has been observed in karyotype data in the 80s and 90s. The authors should really consider reviewing and looking into our understanding of somatic copy number from spectral karyotyping methods.

We have now added discussion of SKY approaches to the introduction.

For example, somatic mosaicism among lymphocytes has been recognized since the 1970's with the discovery of somatic gene rearrangement at T cell receptor and immunoglobulin loci¹. In the late 90's, advances such as spectral karyotyping (SKY)² and multiplex fluorescence in situ hybridization (FISH)³ began to comprehensively map aneuploidy and chromosomal translocations in metaphase spreads from cancer cells. These approaches identified recurrent chromosomal translocations in proliferative cancer cells⁴ leading, in part, to the identification of genomic fragile sites that underlie the ontogeny of many cancers⁵. SKY was also used to identify aneuploidy in acutely isolated mouse neural progenitor cells^{6,7}. And FISH approaches detected aneuploid neurons in mouse and human cerebral cortex^{8,9}. Recent advances in single cell and bulk DNA sequencing approaches have revealed abundant somatic mosaicism throughout the human body¹⁰⁻¹⁴.

Reviewer #3 (Remarks to the Author):

Title: Mapping the complex genetic landscape of human neurons

In this study, the authors employed a droplet-based WGA approach with an allele-based validation to identify 226 CNV neurons out of 2,125 cortical neurons from a neurotypical human brain. Furthermore, they found that CNV location appears to be non-random. Instead, there were CNV hotspots and cold spots in the genome. Recurrent CNV regions contained fewer, but longer genes. The issue of developing a method which accurately identifies single-cell CNVs is very important in the field of somatic mosaicism. In this regard, this study is important and interesting. However, this reviewer is not fully convinced by the result and conclusion of this study due to a couple of reasons. Firstly, as the authors pointed out, WGA events can result in a lot of artifactual CNVs. Although they rigorously applied SCOVAL for the identification of true CNVs in a single cell, their result of CNV hotspots and cold spots should be validated in the different WGA methods such as PTA. Secondly, they sequenced neurons from one person and generalized their finding to human neurons. Again, their result should be validated in other individuals. Taken together, in order to agree with the author's conclusion, this reviewer would like to suggest the use of another WGA method (e.g PTA) in a different cohort.

We thank the reviewer for their comments and appreciate their critical assessment of our work. Indeed, Reviewer #2 had similar concerns, and we have now conducted additional analyses to include additional and independent support for our findings from previous lower-resolution studies using an aggregate of 867 neurons from 15 individuals (please see above).

Other comments;

1. Figure 2 could be more informative. Addition of labelling the specific neurons (specifically the monosomic neurons) in Figure 2 A/B would aid in understanding.

We thank the reviewer for their feedback. We have now improved the figure by including arrows that specifically indicate the monosomic neurons in Figure 2A, B, and C.

2. Clarification of Figure 2C - "monosomic neurons harboured deletions affecting >40% of other chromosomes" - but I cannot see this in the figure.

We thank the reviewer for their comment. We have now constructed a new supplementary figure (Supp Fig. 6A) that explicitly depicts the increased CNV load within monosomic neurons and referenced this in the main text.

All monosomic neuronal genomes were highly aberrant and harbored many additional deletions affecting 40 – 98% of other chromosomes (Fig. 2C, Supplementary Fig. 6A).

A

3. In figure 2D, since the CNVs are apparently not clonal, is the tree informative?

We thank the reviewer for this comment. We used hierarchical clustering to investigate clonality, and such clustered data is typically represented as a tree. While it is true that the CNVs are not apparently clonal, we were able to make this determination by using the analysis presented in Fig 2D. Our expectations were that clonal CNVs will have divergence scores near 0, as can be seen by the pair of sequencing libraries (Cells 17 and 19) with identical karyotypes that we have conservatively interpreted as technical replicates. The tree also allowed us to identify pairs of cells which were seemingly related but which haplotype analysis indicated recurrence at common regions (e.g. Cells 32 and 33). We feel that the figure is an appropriate representation of our analysis.

4. Better explanation and experimentation on how they can combat CNV artifacts - since the conclusions drawn rely on this.

We thank the reviewer for this comment. The inclusion of phased haplotype information in SCOVAL enabled us to remove many (~25%) of our initial CNV calls ascertained by read depth alone which were likely due to uneven WGA in various regions of the genome. We also detail our use of mean absolute deviation (MAD) analysis of individual bins across all cells to also exclude problematic genome regions.

In our response to Reviewer #2, we have now included additional complementary analysis showing consistency with prior results. While this does not exclude CNV artifacts at the individual CNV level, it does provide confidence that

our overall observations on CNV prevalence are likely bona fide and that any such artifacts remaining after our processing and QC are likely minimal.

5. Can they be 100% certain that germline CNVs were not included? Germline CNVs should be detected in many of single neurons.

While we cannot claim to be 100% certain that there were no germline CNVs included in our analysis, our detection resolution of >1Mbp means that such events in the germline would presumably be pathogenic and thus are very unlikely given that the donor was healthy at time of death. In addition, CNVs would indeed show signals in many of the individual neurons, however depending on their size these would, in most cases, be filtered out as bad bins which are labeled as systematic read-depth outliers across all neurons (the median read-depth of bins is used, and bins are filtered using the interquartile range, as noted in the Methods section).

Nevertheless, we have now investigated this further through two different avenues. First, we examined the 10X Linked-Read data from the bulk tissue and called CNVs using LongRanger and Manta. We did not observe any events larger than 1 Mbp or which had any considerable overlap with our somatic CNVs. Next, we examined the minor allele frequencies of heterozygous SNPs across all cells within the coordinates of our somatic CNVs and observed a median minor allele frequency (MAF) ranging from 0.45-0.49 as shown in a new Supplementary Figure 2D, consistent with typical diploid regions. In aggregate, these results coupled with pathogenicity of such large CNVs suggest that the presence of germline CNVs in our somatic set is extremely unlikely. We have now added these analyses to the manuscript.

We next assessed whether any of our somatic CNVs could potentially represent germline variants which escaped our analytical filters. We did this by first examining the 10X Linked-Read data and called CNVs using LongRanger and Manta (Methods). We did not observe any events larger than 1 Mbp or which had any considerable overlap with our somatic CNVs. Next, we examined the minor allele frequencies of heterozygous SNPs across all cells within the coordinates of our somatic CNVs and observed a median minor allele frequency (MAF) ranging from 0.45-0.49 (Supplementary Fig. 2D), consistent with typical diploid regions. Additionally, our detection resolution of >1Mbp suggests that such events in the germline would presumably be pathogenic and thus are unlikely given that the donor was healthy at time of death. In aggregate, these results coupled with pathogenicity of such large CNVs suggest that the presence of germline CNVs in our somatic set is extremely unlikely.

Germline CNV Assessment

To determine whether any potential germline CNVs were included in our analysis, we analyzed the 10X linked-read sequences using both Manta and Long Ranger (<https://github.com/10XGenomics/longranger>) using default settings and compared the detected CNVs with our somatic CNVs using a 50% reciprocal overlap criteria. We further examined the minor allele frequencies of heterozygous SNPs across all cells within the boundaries of detected somatic CNVs with the expectation that germline deletions would have a consistent deviation from .50 frequency if present.

6. The authors should mention how they stain Neun and the FANS plot is needed in the supple figures.

The FANS plot is now included in Supplementary Fig. 1A.

We isolated neuronal nuclei from post-mortem frontal cortex of a 49 year-old, male, neurotypical control by fluorescence-activated nuclei sorting. Using NeuN-positive nuclei (**Supplementary Fig. 1A**), two DNA libraries were prepared in separate lanes on the 10X Genomics Chromium platform (**Fig. 1A**); each lane obtained produced ~1,000 single neuronal genomic libraries with unique barcodes.

Reviewer #1 (Remarks to the Author):

The authors addressed my comments in full, thank you.

Geoff Faulkner (University of Queensland)

Reviewer #2 (Remarks to the Author):

Sun and colleagues examine the acquisition of somatic mutations CNVs in neurons by analysis of alleles through loss of heterozygosity from 2,125 frontal cortical nuclei on a neurotypical individual. This method differs from prior methods by leveraging 10X Genomics single-cell CNV. As the reviewer understands it, this method is no longer supported, lending to the additional value of this dataset.

The reviewer has long suspected that somatic acquisition of copy numbers and other events later in life may play a role in some neurodegenerative disorders. Some early evidence has suggested this as much as years ago, but the data could be clearer at best at any such conclusions. This paper lays a few simple foundations for identifying hotspots within a single individual and only extends a few steps further.

The authors have addressed the concerns I had reasonably well. The only minor critique is that the title "Mapping the Complex Genetic Landscape of Human Neurons" needs to be narrower to help future papers understand the findings of this paper. It would get more future citations if it referred to CNVs in the title.

Reviewer #3 (Remarks to the Author):

The authors addressed this reviewer's concerns well by performing additional analysis in independent cohorts. This reviewer agrees to that this study provides a new insight of CNV neurons in human frontal cortex.

RE: NCOMMS-23-09510B
RE: Response to Reviewer's

March 24, 2023

REVIEWERS' COMMENTS

Reviewer #1 (Remarks to the Author):

The authors addressed my comments in full, thank you.

Geoff Faulkner (University of Queensland)

Reviewer #2 (Remarks to the Author):

Sun and colleagues examine the acquisition of somatic mutations CNVs in neurons by analysis of alleles through loss of heterozygosity from 2,125 frontal cortical nuclei on a neurotypical individual. This method differs from prior methods by leveraging 10X Genomics single-cell CNV. As the reviewer understands it, this method is no longer supported, lending to the additional value of this dataset.

The reviewer has long suspected that somatic acquisition of copy numbers and other events later in life may play a role in some neurodegenerative disorders. Some early evidence has suggested this as much as years ago, but the data could be clearer at best at any such conclusions. This paper lays a few simple foundations for identifying hotspots within a single individual and only extends a few steps further.

The authors have addressed the concerns I had reasonably well. The only minor critique is that the title "Mapping the Complex Genetic Landscape of Human Neurons" needs to be narrower to help future papers understand the findings of this paper. It would get more future citations if it referred to CNVs in the title.

We thank the reviewer for this suggestion. The manuscript is now titled, "**Mapping Recurrent Mosaic Copy Number Variation in Human Neurons.**"

Reviewer #3 (Remarks to the Author):

The authors addressed this reviewer's concerns well by performing additional analysis in independent cohorts. This reviewer agrees to that this study provides a new insight of CNV neurons in human frontal cortex.